# The Medication Safety Adventure Trail: An Educational Intervention to Promote Public Awareness on Medication Safety

**DOI:** 10.3390/pharmacy13030075

**Published:** 2025-05-27

**Authors:** Audrey Flornoy-Guédon, Liliane Gschwind, Antoine Poncet, Pierre Chopard, Caroline Fonzo-Christe, Pascal Bonnabry

**Affiliations:** 1Pharmacy, Geneva University Hospitals, Rue Gabrielle-Perret-Gentil 4, 1205 Geneva, Switzerland; liliane.gschwind@hug.ch (L.G.); caroline.fonzo-christe@hug.ch (C.F.-C.); pascal.bonnabry@hug.ch (P.B.); 2Department of Epidemiology, Geneva University Hospitals, 1205 Geneva, Switzerland; antoine.poncet@hug.ch; 3Chopard Quality Academy in Healthcare, 1202 Geneva, Switzerland; pichopard@gmail.com; 4Institute of Pharmaceutical Sciences of Western Switzerland (ISPSO), School of Pharmaceutical Sciences, University of Geneva, 1205 Geneva, Switzerland

**Keywords:** patient safety, patient empowerment, medication safety, drug utilization, educational tool, serious game, experiential learning

## Abstract

Engaging patients in medication safety is essential but remains under-addressed in hospital settings. This pilot study aimed to assess the impact of an educational intervention—the Medication Safety Adventure Trail—on medication safety knowledge and satisfaction among hospital visitors. A quasi-experimental pre-post intervention using this educational tool was conducted over five days. A booth was set up in a hospital lobby inviting all passers-by to follow a six-step trail involving riddles to solve. The experiment comprised three phases: 1. Briefing plus pre-test; 2. The trail; 3. Debriefing plus post-test. A logistic mixed-effects model was employed to assess changes in the odds of correct responses to eight items between the pre-test and post-test. A five-point scale assessed participants’ degrees of certainty (DC) in their answers, and a comparison pre- and post-test was performed with a linear mixed-effects model. Satisfaction was based on Kirkpatrick’s levels 1 and 2 (reaction and learning) and was assessed using categorical scales and open-ended questions. A total of 93 participants completed the trail (60% non-healthcare professionals, 36% healthcare professionals, and 4% unspecified). The odds of a correct answer were higher at post-test than at pre-test (72% vs. 51%, *p* < 0.001), and the odds of providing a correct answer were nearly five times higher following the activity compared to before (OR = 4.8 [95%CI 3.5 to 6.4], *p* < 0.001). The mean DC was also improved from pre-test to post-test (4.43, 95%CI [4.36–4.49] vs. 4.83, 95%CI [4.80–4.86]; *p* < 0.001). All 93 participants reported being either very satisfied (89%) or satisfied (11%) with the educational tool. The tool significantly improved participants’ knowledge about medication safety issues and was appreciated.

## 1. Introduction

The WHO’s third Global Patient Safety Challenge—Medication Without Harm—promotes the idea that “everyone has a role to play in medication safety”, including patients [1]. The campaign aims to increase public awareness of medication safety issues and calls on patients to “Know. Check. Ask.” [2].

Ensuring medication safety is a critical concern for healthcare professionals throughout the continuum of care, from hospital admission to discharge. Different technologies help to reduce errors from medication prescription to administration. These include computerized prescription systems, automated dispensing cabinets, identification bracelets, and electronic health records [3,4]. Good working practices should ensure that the right patient receives the right drug at the right time in the right dose and by the right route [5]. International accreditation frameworks, such as those from the Joint Commission International (JCI), have also been shown to enhance institutional medication safety practices and staff awareness, supporting the importance of structured efforts in promoting patient safety culture [6]. But do patients know about all these safety measures, and do they know their own role? A study of patient perceptions highlighted that communication about medication during hospital stays warranted improvement [7]. Patients today are taking on ever more responsibility for their own care and are considered essential partners in patient safety [8]. Moreover, patient engagement is associated with patient satisfaction and better care outcomes [9], and it appears to be useful when addressing the challenges of improving healthcare quality [10]. The WHO promotes patient empowerment [8] and recommends awareness-raising interventions to reduce medication-related harm [1].

Using serious games as an awareness-raising intervention is an interesting approach with which to foster active learning [11]. Games generate learner commitment, boost motivation [12], and improve memory and information retention [13]. Several studies have evaluated the impact of serious games on patients’ rehabilitation [14], therapeutic education [15], treatment self-management [16], and medication adherence [17]. Only a few studies exist on serious games for health incorporating medication use, and these games have usually been in digital form [18].

To raise patients’ awareness and promote their empowerment regarding medication safety in hospitals, an engaging, educational tool called the Medication Safety Adventure Trail was created, which supported the principles of the WHO’s slogan [2]. The present study’s objectives were to evaluate the tool’s impact on knowledge about medication safety issues and collect information about participant satisfaction. To the best of our knowledge, no studies have been conducted to evaluate such educational tools for promoting patient empowerment and contributing to their safety from hospital admission to discharge.

## 2. Materials and Methods

### 2.1. Setting

As part of Switzerland’s national patient safety week, our pilot study was carried out during five days in the lobby of a university hospital. Any French-speaking adult entering or staying in the hospital could participate voluntarily, including healthcare professionals—because all individuals are potential recipients of care.

### 2.2. Tool Design

An interprofessional steering committee (two clinical pharmacists, a clinical nurse specialized in drug therapy, a physician, and a nurse from the Quality of Care Unit) defined six safety topics and their related key messages (Table 1).

The format chosen for the tool used the metaphor of the patient’s care pathway during hospitalization with elements of gamification. An external company was hired to design a booth with luminous walls to display the trail instructions on one side and key safety messages on the other.

A volunteer patient was involved in the tool’s development and feasibility testing in order to ensure that the language was appropriate and easy to understand. The Patient Safety Switzerland Foundation provided participants with brochures on medication safety topics and treatment record cards [19].

### 2.3. Implementation of the Educational Tool

The Medication Safety Adventure Trail was divided into three steps: 1. A briefing plus a pre-test questionnaire; 2. Following the adventure trail itself; 3. A debriefing plus a post-test questionnaire. Each phase took place in a different physical location (Figure 1):A briefing area booth dedicated to welcoming participants and explaining the experiment;A trail area recreating the environment of a patient’s room through which participants were guided using a rope and carabiner; along the trail, six steps from hospital admission to discharge were represented by interactive, playful content; at each step, participants had to solve an enigma about safe medication use and write the answer on their roadmap (Appendix A);Two debriefing areas where participants met a clinical pharmacist and/or a clinical nurse specialist after completing the trail to receive the correct answers to the quiz, ask any questions, and review the key safety messages.

The healthcare professionals staffing the trail underwent a training session and received an accompanying booklet to ensure that all the messages given during briefings and debriefings were consistent. A team of ten clinical pharmacists, five clinical specialist nurses, and one scientific officer from the Patient Safety Switzerland Foundation took turns in the booth as participant–debriefers, with four of them always in place.

The approximate cost of setting up the activity was CHF 10,000 (material, communication).

### 2.4. Evaluation of the Tool

The evaluation was based on the first two levels of Kirkpatrick’s training program evaluation model, i.e., reaction and learning [20]. Each participant filled out two structured paper questionnaires: a pre-test questionnaire before the trail collected demographic data (age, sex, treatment, profession, and history of hospitalization) and assessed basic knowledge (Appendix B); a post-test questionnaire, after their debriefing, retested their knowledge (Kirkpatrick’s level 2: learning) and assessed their satisfaction with the trail (Kirkpatrick’s level 1: reaction). All questionnaires were written in French.

### 2.5. Knowledge and Degree of Certainty Assessment

Knowledge testing involved identical eight-question pre-test and post-test multiple-choice questionnaires (Table 2). The differences between the pre-test and post-test scores were used to assess participants’ learning gain (Kirkpatrick’s level 2). The questionnaire was reviewed and pilot-tested by six clinical pharmacists and a volunteer patient to ensure the clarity of the questions and their alignment with the pedagogical activity.

Each question had several potentially correct answers among the options proposed. Question scores were binary: a score of 1 indicating a fully correct answer and 0 otherwise. Omission of any correct option within a given question resulted in a score of zero. The total maximum score attainable was 8 points, and the minimum was 0.

The participant’s degree of certainty (DC) was assessed for each item by answering the question “How sure are you about your answer?” A validated five-point Likert-type scale was used to score participants’ confidence (5–20%: not at all sure; 20–40%: unsure; 40–60%: half sure; 60–80%: quite sure; 80–95%: almost certain) [21]. To express the DC results, a score for each point of the scale was assigned, from 1 = “not at all sure” to 5 = “almost certain”.

### 2.6. Satisfaction Assessment

At the end of the experiment, participants’ reactions (Kirkpatrick’s level 1) were evaluated using an additional seven-item questionnaire (Appendix C) to measure satisfaction with the trail. Two items were single-choice questions about the usefulness of the information provided and its level of difficulty. Two questions assessed participants’ satisfaction with the concept and the teaching method using a four-step smiley scale. Three open-ended questions invited participants to describe what they enjoyed most and least and to leave a comment. This questionnaire was also pre-tested by six clinical pharmacists and a volunteer patient to ensure the clarity.

### 2.7. Statistical Analysis

Individuals’ questionnaire responses were transferred into an Excel file, and statistical analyses were performed using StatXact 8 (Cytel Inc., Cambridge, MA, USA) or Stata/IC 16 (StataCorp., College Station, TX, USA: StataCorp LLC). Participant demographics were described as counts and percentages for sex, profession, current drug treatment, and previous hospitalizations, and as mean and range for age.

Improvements in knowledge were evaluated by comparing the probability (odds) of a correct answer between the pre-test and post-test using a logistic mixed-effects model with two-way crossed random effects on participant and question. The number of correct answers between pre- and post-test was also compared using a paired Student test. Improvements in participants’ degrees of certainty were evaluated with a linear mixed-effects model with two-way crossed random effects on participant and question. Subsequently, a question-by-question analysis was conducted to compare the rates of correct answers between pre- and post-test using McNemar’s tests. Finally, and for both occasions (pre- and post-test), the mean number of correct answers was compared between healthcare professionals and other participants using a Student test. Statistical significance was set at the two-sided 0.05 level for all analyses.

## 3. Results

### 3.1. Study Population

One hundred people passing through the hospital lobby volunteered to participate (mean age = 48 years old [range 18–88], 41% men): 60% were not healthcare professionals (36% were healthcare professionals, and 4% did not communicate their profession), 56% were receiving current drug treatment, and 85% had previously been hospitalized. Ninety-three participants completed all the questionnaires and were included in the pre-test/post-test analysis and the comparison between healthcare professionals (n = 35) and those who were not (n = 56). Two persons did not communicate their profession.

### 3.2. Knowledge and Degree of Certainty

#### 3.2.1. Comparison Between Pre-Tests and Post-Tests for All Participants

To compare the odds of a correct answer between pre-test and post-test, a logistic mixed-effects model with two-way crossed random effects on participant and question was used. The probability of a correct answer increased significantly from pre-test (51%, n = 744) to post-test (72.4%, n = 744; *p* < 0.001). The odds of providing a correct answer were nearly five times higher following the activity compared to before (OR = 4.8 [95%CI 3.5 to 6.4], *p* < 0.001). The number of correct answers between the pre- and post-test was also compared using a paired Student’s t-test. On average, participants correctly answered 4.1 ± 2.2 questions in the pre-test and 5.8 ± 2.1 questions in the post-test (mean difference = 1.7 [95%CI 1.3 to 2.1], *p* < 0.001).

Improvements in participants’ degrees of certainty were evaluated with a linear mixed-effects model with two-way crossed random effects on participant and question. The participants’ DC also increased significantly from pre-test to post-test (respectively, 4.43, 95%CI [4.36–4.49] vs. 4.83, 95%CI [4.80–4.86]; *p* < 0.001).

The question-by-question analysis comparing the rates of correct answers between pre- and post-test was performed using McNemar’s tests. Participants’ pre-test knowledge levels were mostly low or moderate across the different questions (i.e., a maximum of 68% of correct answers). However, the question “Do you think you have a role to play in your medication management in hospital?” was answered correctly by 87% of participants at baseline (Figure 2). The rate of correct answers rose significantly for all questions in the post-test. The three questions showing the most improvement in the post-test were “If you cannot swallow a tablet (e.g., it is too big), what do you do?” (+27%), “In hospital, who is involved in checking that you receive the right drug, at the right time, in the right dose and by the right route?” (+30%) and “When you are discharged from hospital, some of your medications have been changed and you have questions; who can answer them?” (+27%).

#### 3.2.2. Comparison Between Healthcare Professionals and Other Participants

Among the 93 participants who completed the experience, two did not communicate their profession. In the remaining 91 participants, significant improvements in knowledge were observed regardless of profession (Table 3).

Degrees of certainty also improved among healthcare professionals (from 4.57 [95%CI 4.48–4.66] in the pre-test to 4.88 [4.84–4.92] in the post-test; *p* < 0.001) and others (from 4.36 [4.27–4.45] to 4.79 [4.75–4.83]; *p* < 0.001; linear mixed-effects model).

For the pre- and post-tests, the mean number of correct answers was compared between healthcare professionals and other participants using a Student t-test. Rates of correct answers were higher among healthcare professionals at both pre-test (66.3% vs. 42.5%; *p* < 0.001) and post-test (81.3% vs. 67.5%; *p* = 0.010). Indeed, at post-test, other participants had reached the mean number of correct answers that healthcare professionals had attained in the pre-test (Table 3).

The question-by-question analysis (McNemar’s tests) showed a significant increase in the rate of correct answers for all post-test questions among participants who were not healthcare professionals, except for the question about drug record cards. Knowledge at baseline was highest for the question about the patient’s role (84%). The question that exhibited the most improvement at post-test was about not being able to swallow a tablet (+34%).

Analysis of healthcare professionals showed improvements on all post-test questions, but the differences between pre-test and post-test were only statistically significant for “What is a drug record card used for?” and “In hospital, who is involved in checking that you receive the right drug, at the right time, in the right dose and by the right route?”. The highest pre-test score was for “When you are discharged from hospital, some of your medications have been changed and you have questions; who can answer them?”. The question that exhibited the greatest improvement at post-test was “In hospital, who is involved in checking that you receive the right drug, at the right time, in the right dose and by the right route?” (+31%).

### 3.3. Satisfaction

All 93 participants reported being either very satisfied (89% with the educational tool and 85% with their participation experience) or satisfied (11% and 15%, respectively). The educational tool’s level of difficulty was judged appropriate by 92% of the participants; none found the trail game too difficult, but seven (8%, including five healthcare professionals) found it too easy. In their comments, participants reported a high degree of satisfaction with the tool’s playful aspects and the pillbox challenge (22 and 7 times, respectively). Among the few negative comments, five participants did not appreciate “being tied” to the rope simulating a via ferrata, one person found the trail game “a little bit long”, and one thought that the mannequin in the bed was “scary”.

## 4. Discussion

### 4.1. Summary of Results

The engaging, educational tool—the Medication Safety Adventure Trail—significantly improved participants’ knowledge and confidence about medication use safety in the hospital from admission to discharge. All the participants were satisfied, found the level of difficulty appropriate, and stated that the information provided was useful. The concept and educational materials were validated and could be used again at other sites or on other occasions. The intervention produced statistically significant improvements in medication safety knowledge and confidence, suggesting its utility as an educational strategy aimed at the public, with potential for broader implementation in hospital-based awareness programs.

### 4.2. The Tool’s Impact on Promoting Empowerment in Medication Safety

This educational tool fostered patient empowerment through two primary mechanisms: enhancing awareness of essential elements required to ensure medication safety and reinforcing the patient’s role as an active participant in their own treatment process. As the WHO has pointed out, it is crucial for health system users to “Know. Check. Ask.” about medication issues [2] in order to become partners in patient safety [7], including during their hospital stay. This tool supported each element of this framework.

“Know”: Results from the pre-test showed that although 85% of participants had previously been hospitalized and 56% were taking a current drug treatment, many of them were unfamiliar with the safety measures implemented to ensure best practice medication management in hospitals. This tool improved knowledge on different safety issues: utility of drug record cards (+22% of correct answers), contribution of computerized medical data (+18%), what to do with tablets that cannot be swallowed (+27%), value of ID bracelets (+23%), and utility of pillboxes (+15%).

The study highlighted a surprising lack of awareness about the usefulness of drug record cards (49% of correct answers in the pre-test)—also known as medication plans—that list all the current medications taken and treatments being followed by a patient. These are particularly useful at hospital admission for ensuring the continuity of medication prescriptions during hospitalization [22,23], and they must be updated at hospital discharge. A detailed medication plan can also help when preparing a pillbox and organizing a patient’s drug administration. Indeed, when patients know about their drug record card, they are eager to use it after their discharge and feel more involved in their care [24].

The omnipresence of computers in hospitals may seem intrusive [25], and this affects relationships between healthcare professionals and patients [26]. However, computerized systems contribute to safer medical prescriptions by centralizing data and facilitating access to them by different healthcare professionals. For this to be better accepted, it is probably important that hospital users understand its benefits.

“Check”: Patients were interested in learning about how the whole medication process was organized. Knowing that medication safety issues exist is the first step towards checking that protocols or best practices are being followed, which then allows patients to contribute to their own safety in the hospital and at home. In addition, the trail emphasized key safety messages, such as the importance of verifying expiration dates and ensuring the correct medication is taken at the appropriate dose and time.

The patient’s key role in their own medication management was known by all the participants (87% of correct answers in the pre-test). The experiment seemed to reinforce the notion that everyone can and should be a vigilant partner in their medication management (97% of correct answers in the post-test) and communicate openly with professionals [27] to help avoid medication errors [28].

“Ask”: The literature reveals that avoidable medication administration errors in the home are common [29,30]. These can potentially lead to significant harm or lessen a treatment’s effectiveness [31]. As observed in the pre-test, a sizeable proportion (41%) of participants who were not healthcare professionals did not correctly answer the question about what to do in case of a hard-to-swallow tablet. Inappropriate use of medication, like cutting or crushing a drug wrongly, for example, is likely to increase as the population ages and must often deal with complex polymedication and difficulty swallowing [32]. This educational tool succeeded in raising participants’ awareness about asking healthcare professionals this type of question. Healthcare professionals, such as nurses, doctors, and pharmacists, are all involved in ensuring that the right patient receives the right drug, at the right time, in the right dose, by the right route. Each of these professions are resource persons for patients seeking to ask questions about their drugs during and after hospitalization. However, the results revealed that people who were not healthcare professionals were not sure to whom to turn to. This seems to be an issue requiring improvement in the future.

A range of interventions, implemented in various formats, have been developed to promote the safe and effective use of medications among consumers. Although isolated, one-time interventions have shown limited effectiveness in enhancing medication adherence or clinical outcomes, they have demonstrated utility in improving patients’ knowledge and raising awareness regarding medication use [33].

### 4.3. Strengths and Limitations

To the best of our knowledge, the Medication Safety Adventure Trail is the first patient-adapted tool to promote public awareness on the safety issues related to medication use in hospitals. Other entertaining room of errors or escape game teaching tools have previously been used to increase patient safety, but they were developed for healthcare professionals [34,35,36,37,38,39,40]. This tool’s positive impact on knowledge and participants’ high levels of satisfaction with it showed it to be an effective means of conducting a public awareness intervention.

The present study had some limitations. The educational intervention was conducted at a single site, which may limit generalizability. A selection bias may have occurred due to the voluntary participation, and the absence of a control group limited causal interference. The tool was limited to the French-speaking participants, reducing cultural and linguistic applicability.

The increase in knowledge among non-healthcare participants was notable. However, in the post-activity survey, their accuracy remained at or below 75% on six of the eight questions. While the activity appeared to enhance medication safety awareness within this subgroup, the overall impact was moderate. The absence of formal psychometric validation of the questionnaires may affect the reliability of the measurements obtained. The concept of having more than one possible answer seemed difficult for some people to grasp and probably introduced some bias into our results. For example, for question 7, “In hospital, who is involved in checking that you receive the right drug, at the right time, in the right dose and by the right route?”, all of the available answers were valid, but few participants ticked all the boxes correctly, even among the healthcare professionals.

Another limitation was that the experiment required many staff: with a minimum of three on hand at all times, this was a time-consuming exercise. It should also be pointed out that setting up this activity came at a significant cost.

Outcomes were assessed only in the short term, without evaluating long-term retention.

### 4.4. Future Research

To address some of the present study’s limitations, it would be interesting to retest the participants’ knowledge several months after the experiment to measure knowledge retention over time. Future research could involve conducting this study in additional sites to evaluate the broader applicability of the results.

Although this educational tool was designed for patients, healthcare professionals also participated and demonstrated higher baseline knowledge. Understandably, they obtained better scores than other participants and fully understood certain topics (ID bracelets, solutions involving cutting or crushing tablets). Their post-test results showed more correct answers for every question but statistically significant improvements for only two (Q1 on drug record cards and Q6 on whether patients have a role in medication management). It would be of interest to develop a similar educational tool specifically designed for healthcare professionals and best suited to their daily practice and level of knowledge.

This approach could also be adapted to develop similar tools aimed at raising awareness around other healthcare-related issues.

## 5. Conclusions

The present quasi-experimental intervention demonstrated that the Medication Safety Adventure Trail significantly improved participants’ knowledge and confidence about medication use safety in the hospital from admission to discharge. Furthermore, both healthcare professionals and other participants reported a high degree of satisfaction with the educational intervention, highlighting the value of the integration of playful elements. It was a suitable educational tool for successfully increasing public awareness of medication safety and promoting patient empowerment. It would be interesting to conduct multi-site studies to strengthen the quality of the evidence presented here, and future research could also assess the long-term impact of such an intervention.

## Figures and Tables

**Figure 1 pharmacy-13-00075-f001:**
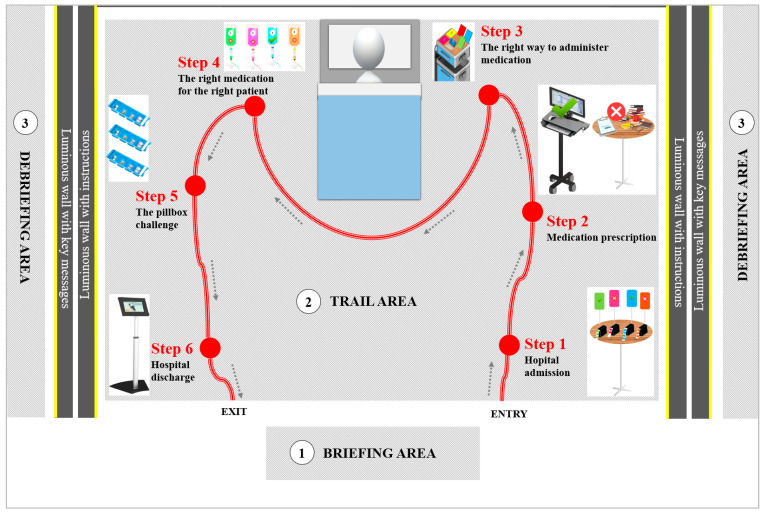
Schematic illustration of the educational tool “Adventure medication safety trail”.

**Figure 2 pharmacy-13-00075-f002:**
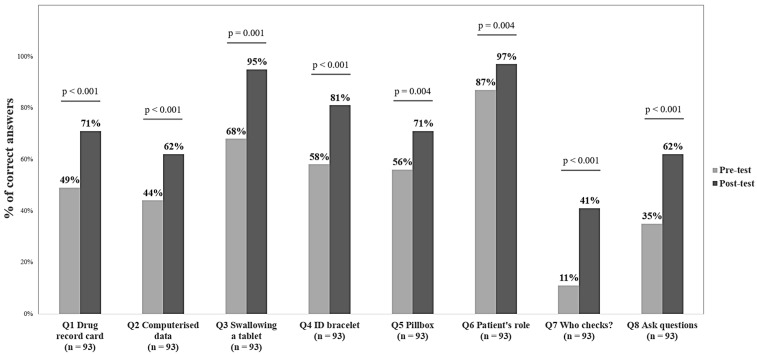
Percentages of correct answers before (pre-test) and after (post-test) Medication Safety Adventure Trail for all participants (n = 93).

**Table 1 pharmacy-13-00075-t001:** Safety topics and key messages.

Safety Topics	Key Safety Messages
Admission to hospital	Utility of drug record cards for listing all drugs and treatments Importance of communicating the entire list of medication one takes
Medical prescription	Value of computerized prescription systems in ensuring patient safety
The right way to administer medication	Good practices in medication administration (e.g., cut or crush a tablet) Multidisciplinary approach to medication management (doctor, nurse, pharmacist, patient)
The right medication for the right patient	Utility of identification bracelets Importance of matching prescriptions, medications administered, and patient identities
The pillbox challenge	Utility of a medication plan for preparing pillboxes Importance of checking expiry dates and taking the right drug at the right dosage at the right time
Hospital discharge	Importance of asking different healthcare professionals questions about one’s treatment Importance of passing on key messages about safe medication use at home Value of having an electronic health record

**Table 2 pharmacy-13-00075-t002:** Eight knowledge assessment questions.

Q1. What is a drug record card used for?
Q2. Why do you think your medical data are stored on a computer when you are in hospital?
Q3. If you cannot swallow a tablet (e.g., it is too big), what do you do?
Q4. Why do you have to wear an identification (ID) bracelet during your hospitalisation?
Q5. What is a pillbox for?
Q6. Do you think you have a role to play in your medication management in hospital?
Q7. Who in the hospital is involved in checking that you receive the right drug, at the right time, in the right dose and by the right route?
Q8. When you are discharged from hospital, some of your medications have been changed and you have questions; who can answer them?

**Table 3 pharmacy-13-00075-t003:** Pre-test and post-test answers among healthcare professionals and other participants.

Questions	Healthcare Professionals * (n = 35)			Other Participants * (n = 56)		
	Pre-Test % (n)	Post-Test % (n)	*p* Value	Pre-Test % (n)	Post-Test % (n)	*p* Value
Q1. Drug record card	66 (23)	89 (31)	0.008	41 (23)	59 (33)	0.064
Q2. Computerised data	60 (21)	71 (25)	0.219	36 (20)	57 (32)	0.008
Q3. Can’t swallow tablet	83 (29)	97 (34)	0.125	59 (33)	93 (52)	<0.001
Q4. ID bracelet	86 (30)	89 (31)	1	43 (24)	75 (42)	<0.001
Q5. Pillbox	63 (22)	71 (25)	0.453	52 (29)	71 (40)	0.007
Q6. Patient’s role	57 (20)	74 (26)	0.109	84 (47)	95 (53)	0.031
Q7. Who checks?	23 (8)	54 (19)	0.001	4 (2)	32 (18)	<0.001
Q8. Ask questions	94 (33)	100 (35)	0.500	23 (13)	54 (30)	<0.001
**Number of correct** **answers, mean (SD)**	5.31 (1.94)	6.46 (1.75)	<0.001	3.41 (1.92)	5.36 (2.19)	<0.001

* Two participants: unknown profession.

## Data Availability

The datasets used and/or analysed during the current study are available from the corresponding author on reasonable request.

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
