# Peer review of "The Medication Safety Adventure Trail: An Educational Intervention to Promote Public Awareness on Medication Safety"

_pharmacy, 2025, doi:10.3390/pharmacy13030075_

Round 1

Reviewer 1 Report

Comments and Suggestions for Authors

As per the attached PDF review report 

Comments on the Quality of English Language

As per the PDF review report

Reviewer 2 Report

Comments and Suggestions for Authors

This study explains the deployment of and results from a newly developed Medication Safety Adventure Trail. Overall, this is an interesting and novel study. However, the authors can revise the work for more clarity and transparency.

A few questions for the authors regarding the manuscript:

Methods

  • What was the average time taken by participants to complete the trail?
  • What was the approximate cost incurred to set up the activity? 
  • What is meant by this phrase? Failing to select a single correct proposal for a question led to a score of zero - does this mean that even if one of several possible answers was selected, the response was scored as 1?
    • The authors should rephrase this sentence to offer clarity on scoring
  • Were the questions/instructions framed in French or English for the participants? The methods emphasize a French-speaking audience - if the questions/instructions/tests were in English (as provided in the manuscript), how was language interpretation handled for participants who were not well-versed in the English language? 

Limitations/Conclusions

  • The trail was offered once for 5 days at a single location. The "pilot" aspect of this study should be highlighted at relevant places in the manuscript (abstract, methods, conclusion)
  • The jump in knowledge for non-healthcare participants is significant and worth highlighting, as the authors have done. However, in the post-activity survey, the knowledge level in this subgroup stayed at or below 75% in 6 out of 8 questions. The activity did seem to have raised medication safety awareness, but it wasn't astounding for non-healthcare participants. I would therefore suggest highlighting this fact more clearly in the discussion/limitation section of the manuscript.  
